# SEE THE WORLD THROUGH COLOR-TINTED GLASSES FOR BETTER HYPERSPECTRAL RECONSTRUCTION

## ABSTRACT

Hyperspectral reconstruction (HSR) from RGB images is a fundamentally ill-posed problem due to severe spectral information loss. Existing approaches typically rely on a single RGB image, limiting reconstruction accuracy. In this work, we propose a novel multi-image-to-hyperspectral reconstruction (MI-HSR) framework that leverages a triple-camera smartphone system, where two lenses are equipped with carefully selected spectral filters. Our configuration, grounded in theoretical and empirical analysis, enables richer and more diverse spectral representations than conventional single-camera setups. To support this new paradigm, we introduce Doomer, the first dataset for MI-HSR, comprising aligned images from three smartphone cameras and a hyperspectral reference camera across diverse scenes. We show that the proposed HSR model achieves consistent improvements over existing methods on the newly proposed benchmark. In a nutshell, our setup allows 30% towards more accurately estimated spectra compared to an ordinary RGB camera. Our findings suggest that multi-view spectral filtering with commodity hardware can unlock more accurate and practical hyperspectral imaging solutions.

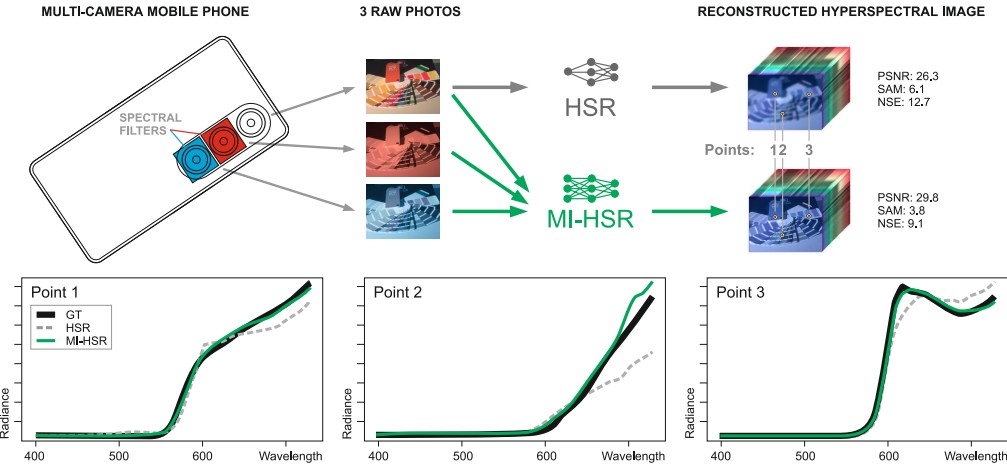

Figure 1: Proposed low-cost mobile spectral imaging system that transforms a standard smartphone into a spectrally diverse capture device via external filters on auxiliary cameras. This configuration enables simultaneous, multi-channel acquisition without internal hardware modification, supporting practical and scalable hyperspectral reconstruction.

## 1 INTRODUCTION

Hyperspectral imaging provide dense spectral measurements at each spatial pixel forming a 3D cube $\mathbf{I}_{HS} \in \mathbb{R}^{h \times w \times n}$, where $n \gg 3$. This enables fine-grained analysis of material properties in applications ranging from remote sensing (Xu et al., 2016), to medical diagnostics (Wang et al., 2023), to historical preservation (Kim et al., 2011), to ISP improvement (Zhou et al., 2024; Glatt

et al., 2024), to food quality assessment (Sun et al., 2024; Ahmed et al., 2024). However, acquiring such high-dimensional data typically requires hardware expensive, bulky, and often reliant on time-consuming scanning thus fundamentally limiting the usability of hyperspectral imaging in dynamic or consumer settings.

An increasingly studied alternative is *hyperspectral reconstruction* (HSR): recovering $\mathbf{I}_{\mathrm{HS}}$ from an RGB observation $\mathbf{I}_{\mathrm{RGB}} \in \mathbb{R}^{h \times w \times 3}$ captured via a sensor with spectral sensitivity matrix $\boldsymbol{S} \in \mathbb{R}^{3 \times n}$ under noise $\mathbf{N}$:

$$\mathbf{I}_{\mathrm{RGB}}(x, y) = \boldsymbol{S} \cdot \mathbf{I}_{\mathrm{HS}}(x, y) + \mathbf{N}(x, y).$$

This inverse problem is highly ill-posed and advances in deep learning-based HSR (Cai et al., 2022; Zhao et al., 2020), reconstructing $\mathbf{I}_{\mathrm{HS}}$ from a single RGB view remains fundamentally limited by the low spectral observability.

Several efforts have aimed to improve this observability. Learned or optimized multispectral filter arrays (MSFAs) (Wu et al., 2019), end-to-end spectral sensitivity learning (Nie et al., 2018), and joint sensor-network co-design (Fu et al., 2018) have all been proposed. However, these methods assume control over sensor hardware and manufacturing thus making them impractical for scalable or consumer-level deployment.

One underexplored but promising path is to leverage the multi-camera systems found in modern smartphones. These devices already include multiple rear cameras with different lenses and spectral sensitivities. In principle, such a setup can be treated as a low-cost, multi-spectral capture system thus capable of observing a scene through multiple, distinct sensitivity matrices. However, prior works on leveraging multiple cameras (Sharma et al., 2023; Oh et al., 2016) do not tackle with unavoidable image misalignment.

*Can we turn a multi-camera smartphone into a compact, spectrally diverse imaging system, without altering its internal hardware?* Our key insight is that by modulating the spectral response of the auxiliary cameras using carefully chosen external filters, we can create an imaging system that allows richer spectral representations. Unlike synthetic MSFAs or custom hardware, our approach requires no internal modification and enables real-time, parallel acquisition. Critically, this setup is easy to deploy, manufacturable at scale, and compatible with existing mobile infrastructure.

We select filters using spectral information loss minimization with respect to a prior hyperspectral distribution. The resulting setup produces spatially misaligned but spectrally rich multi-view data thus posing a new fusion problem that we address via alignment-aware learning. An overview of our physical configuration is shown in Fig. 1, which illustrates how external filters are applied to the auxiliary lenses of a standard smartphone to create a spectrally diverse input set. The combination of low-cost physical augmentation and learning-based reconstruction represents a practical path toward deployable hyperspectral imaging in unconstrained environments. We propose a complete, low-cost pipeline for *multi-image-to-hyperspectral reconstruction (MI-HSR)* using a filter-modified smartphone. Specifically, the contributions are three-fold:

- A novel smartphone-based acquisition system that uses two custom spectral filters over auxiliary cameras, converting a consumer-grade smartphone into a 9-channel imaging device. We analyze and justify our filter choices via information-theoretic criteria. To the best of our knowledge, such a configuration has not been previously explored in HSR literature. Our system significantly outperforms RGB-only and naive multi-view baselines.

- The *Doomer* dataset, the first benchmark for MI-HSR. It contains 4 captures per scene: three from each of smartphone's camera and the fourth from the hyperspectral camera.

- A principled reformulation of transformer-based HSR architecture for our setting, showing that spatial-first attention enables implicit alignment and effective fusion of misaligned inputs across camera viewpoints.

## 2 RELATED WORK

**Low-cost multispectral imaging** Numerous approaches have aimed to capture multispectral or hyperspectral information without expensive hardware. Early work by Helling et al. (2004) employed a grayscale camera with a rotating filter wheel, while Valero et al. (2007) used an RGB camera with three interchangeable filters. Oh et al. (2016) captured scenes with three different cameras, leveraging

the variation in their spectral sensitivities. More recently, Sharma et al. (2023) demonstrated that consumer mobile devices with both RGB and NIR sensors can achieve extended spectral capture (400–1000 nm). Although these systems reduce costs, they either involve long capture times or assume no misalignment between successive captures.

**Hyperspectral reconstruction from RGB** Traditional HSR methods model spectra using sparse coding (Arad & Ben-Shahar, 2016), dictionary learning (Aeschbacher et al., 2017), or manifold embeddings (Jia et al., 2017), based on the low-dimensional structure of hyperspectral data and the rarity of metamers (Foster et al., 2006). These approaches are computationally efficient but often lack the capacity to incorporate global context from the input image, making them less robust to complex natural scenes. Recent methods based on deep learning have achieved significant advances, particularly those developed through the NTIRE spectral reconstruction challenges (Arad et al., 2018; 2020; 2022). Early deep models used CNNs (Zhao et al., 2020; Shi et al., 2018; Xiong et al., 2017; Zhang et al., 2020; Stiebel et al., 2018; Galliani et al., 2017), while newer transformer-based approaches like MST++ (Cai et al., 2022) and MSFN (Wu et al., 2024) introduced attention along spectral or spatial dimensions. However, nearly all research relies on synthetic RGB inputs rendered from hyperspectal images (HSIs) using color matching functions (CMFs), assuming perfect alignment and access to camera parameters. These assumptions do not hold in practice, limiting model generalizability. We address these limitations training and evaluating on real-world data with acquisition artifacts and misalignment.

**Hyperspectral datasets** Several datasets support HSR research. Early datasets like CAVE (Yasuma et al., 2010) and Harvard (Chakrabarti & Zickler, 2011) provided controlled hyperspectral measurements. Later datasets such as ICVL (Arad & Ben-Shahar, 2016), KAUST (Li et al., 2021), and ARAD_1K (Arad et al., 2022) focused on enabling data-driven HSR methods. These datasets contain either radiance or reflectance data, but most lack real RGB images and instead simulate RGB via CMFs, which fails to capture the characteristics of camera pipelines. Moreover, they often assume perfect alignment, which does not hold in practical settings. Recent datasets like BeyondRGB (Glatt et al., 2024) and MobileSpec (Zhou et al., 2024) address some of these issues by including real RGB captures. BeyondRGB includes color charts and lightsource spectrum estimation, while MobileSpec offers aligned RGB-HSI pairs. However, they still face trade-offs between alignment, diversity, and color reference availability.We build on these efforts by introducing Doomer dataset with real RGB images, misaligned hyperspectral data, and in-scene color references thus enabling realistic reconstruction under natural capture conditions.

**Handling misalignment** Misalignment between inputs and ground truth is a well-known challenge in video and reference-based super-resolution tasks. Optical flow (OF) (Kim et al., 2018; Chan et al., 2021), deformable convolutions (Tian et al., 2020; Wang et al., 2019), and attention mechanisms (Wang et al., 2021) have been proposed to mitigate this. In our context, misalignment also arises due to ground truth HSIs being not spatially aligned with the RGB input. Zhang et al. (2021) addressed this by warping ground truth toward the input using OF, enabling pixel-level evaluation. Elezabi et al. (2024) proposed contextual losses and pseudo-aligned inputs as training strategies. We adopt the interpretable and evaluation-friendly approach (Zhang et al., 2021) of warping the ground truth to the input using OF, allowing accurate pixel-wise supervision and metric computation.

## 3 Proposed Imaging System

### 3.1 System Overview

Our goal is to improve HSR by increasing the number of spectrally distinct measurements captured simultaneously, using only consumer-grade hardware. Rather than relying on custom sensor arrays or coded optics, we build upon multi-camera smartphones, which are already equipped with multiple rear-facing cameras featuring different lenses and spectral sensitivity functions (SSFs).

To amplify spectral diversity, we augment the auxiliary cameras with external filters. As illustrated in Fig. 1, this converts each RGB camera into a spectrally modulated sensor. The resulting device captures nine distinct spectral channels: three from each RGB camera, modified by its filter without requiring scanning or internal hardware changes.

| Dataset | # scenes | Spectral data | Color reference | Spectral sampling Range (step), nm | Corresponding RGBs |
|---|---|---|---|---|---|
| CAVE (Yasuma et al., 2010) | 32 | Reflectance | Color chart or no | 400–700 (10) | simulated BMP |
| Harvard (Chakrabarti & Zickler, 2011) | 79 | Radiance | No | 400–720 (10) | No |
| ICVL (Arad & Ben-Shahar, 2016) | 200 | Radiance | Color chart or no | 400–1000 (10) | simulated JPEG |
| KAUST (Li et al., 2021) | 409 | Reflectance | White patch | 400–730 (10) | No |
| ARAD_1K (Arad et al., 2022) | 950 | Radiance | No | 400–700 (10) | simulated JPEG |
| Beyond RGB (Glatt et al., 2024) | 1680 | Radiance | Color chart | 380–730 (20) | 2× real RAW |
| MobileSpec (Zhou et al., 2024) | 200 | Radiance | No | 400–1000 (10) | real RAW |
| Doomer (proposed) | 143 | Radiance | Gray ball | 400–730 (10) | 3× real RAW |

Table 1: Comparison of existing hyperspectral datasets. Our proposed Doomer dataset uniquely offers real multi-view RGB images with spectral filters, misalignment, and in-scene gray reference under diverse conditions.

Formally, let $\boldsymbol{S}_i \in \mathbb{R}^{3 \times n}$ be the SSF of camera $i$, and $\boldsymbol{f}_i \in [0, 1]^n$ the spectral transmittance of the filter applied to that camera. The effective per-camera response becomes $\boldsymbol{S}_i \odot \boldsymbol{f}_i$, and the overall system response is:

$$\boldsymbol{S_F} = \left[ (\boldsymbol{S}_1 \odot \boldsymbol{f}_1)^\top; \ldots; (\boldsymbol{S}_k \odot \boldsymbol{f}_k)^\top \right]^\top \in \mathbb{R}^{3k \times n},$$

where $\boldsymbol{F} = [\boldsymbol{f}_1^\top; \ldots; \boldsymbol{f}_k^\top]$ is the filter configuration and $\odot$ is element-wise multiplication. In our prototype, $k = 3$, with one unfiltered camera ($\boldsymbol{f}_1 = (1, \ldots, 1)^\top$) and two filtered cameras. This design simplifies dataset collection (Sec. 4) while still significantly enriching spectral representations.

This configuration has two practical advantages. First, all channels are captured simultaneously under natural illumination, making it suitable for dynamic scenes. Second, it relies entirely on off-the-shelf hardware components. However, as each camera has a distinct physical position, the resulting images are spatially misaligned. This necessitates learning-based alignment modules, which we incorporate into our reconstruction pipeline (Sec. 5).

### 3.2 FILTER SELECTION VIA SPECTRAL UNCERTAINTY MINIMIZATION

The effectiveness of our imaging system critically depends on the choice of spectral filters. Since we train on fully real-world data, the filter configuration must be fixed prior to data collection.

Given a library of $N = 65$ candidate filters available in our lab, we exhaustively evaluate all $65 \times 64$ ordered filter pairs for the two auxiliary cameras. The optimal pair should minimize spectral ambiguity — i.e., the uncertainty in the latent spectrum $\mathbf{r} \in \mathbb{R}^n$ given a measurement $\mathbf{c} = \boldsymbol{S_F}\mathbf{r} + \mathbf{n}$, where $\boldsymbol{n}$ is sensor noise.

Following Reutskii & Ershov (2024), we use the expected conditional variance of the spectrum as our selection criterion:

$$v(\boldsymbol{F}) = \mathbb{E}_\mathbf{c} \left[ \operatorname{tr} \operatorname{Var}_\mathbf{r}(\mathbf{r} \mid \mathbf{c}) \right].$$

This criterion reflects the average spectral uncertainty remaining after observing $\mathbf{c}$. Lower $v(\boldsymbol{F})$ implies more informative measurements and improved reconstructability.

To compute this metric, we sample spectra $\mathbf{r}$ from a uniform distribution over pixels in a publicly available hyperspectral dataset (Li et al., 2021). We use precise SSFs for our smartphone cameras and spectrophotometer-measured filter transmittances (see Sec. A.2 for details). The final filter pair (Fig. 2) is selected as the one that minimizes $v(\boldsymbol{F})$.

To verify the proposed filter selection strategy, we trained the MI-HSR method in simulated settings with 12 different filter sets. See Sec. A.3 for details.

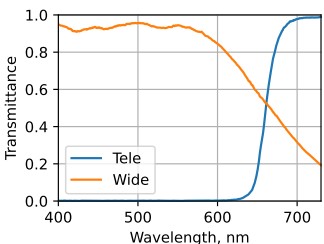

Figure 2: Filters selected for Tele and Wide cameras.

## 4 DOOMER DATASET

**Overview.** To support training and evaluation for the MI-HSR task introduced in Sec. 3, we collect a new dataset, which we call *Doomer*[1]. Existing hyperspectral datasets are not suited for our setup: none provide spatially misaligned multi-view RGB observations. Furthermore, no existing dataset aligns with our specific hardware configuration (a triple-camera smartphone with custom spectral filters), making new data collection a necessity.

Doomer contains 143 real-world scenes captured using a Huawei Mate 40 Pro smartphone equipped with Main, Tele, and Wide cameras, along with a Specim IQ hyperspectral camera for ground truth. In each scene, we record three smartphone RAW images, two with custom spectral filters and one unfiltered as well as a 111-band (400 – 730 nm) hyperspectral image. Example captures are shown in Fig. 3.

Scenes include a mix of indoor and outdoor environments under varying illumination conditions (e.g., halogen and LED lighting indoors; overcast and sunny weather outdoors), and span a range of objects including food, printed material, architectural surfaces, and color calibration charts. Each scene includes a gray ball reference for future work on illumination estimation. While the gray ball is visible in Wide and Main cameras, it falls outside the Tele field of view and is cropped out during preprocessing. Non-preprocessed versions will be included in the public release.

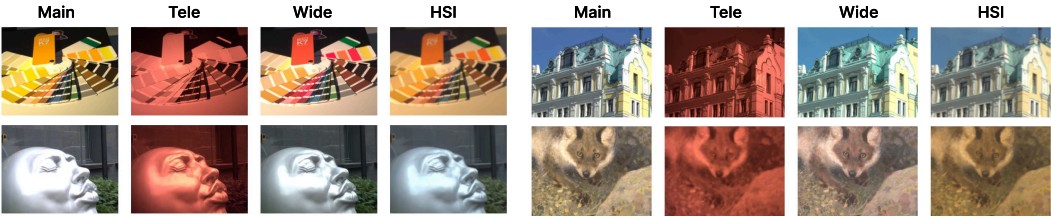

Figure 3: Sample scenes from the Doomer dataset. Smartphone images are rendered to sRGB using device-specific color matrices; hyperspectral images are rendered using CIE RGB CMF.

Gray ball reference makes the proposed dataset potentially useful in research on illumination estimation, automatic white balance and color space transform (see Sec. A.4 for more detailed proposals).

**Capture setup** The data acquisition rig consists of a Specim IQ hyperspectral camera, a Huawei Mate 40 Pro smartphone mounted in a 3D-printed case with slots for spectral filters and a gray reference sphere (VFX ball).

The entire system is mounted on a tripod, with the phone positioned to rotate along a vertical axis for alternating captures (Fig. 4). This design allows the smartphone and hyperspectral camera to image scenes from nearly identical viewpoints, minimizing parallax and occlusion. The gray ball is connected to a rigid rod that allows to regulate the position of the ball in the scene.

Image acquisition proceeds sequentially: first, all three smartphone cameras capture RAW images; then, the smartphone case is moved to make hyperspectral image. Most smartphone settings (e.g., ISO, shutter speed) are controlled automatically, except in scenes with poor red signal where we manually adjusted Tele exposure to avoid excessive noise.

**Preprocessing pipeline** Each four-image scene group (three RGB + one HSI) undergoes standardized preprocessing (see Sec. A.4). We also normalize field of view and resolution across sensors. Specifically, we estimate pairwise homographies between each RGB image and the Tele view using SIFT keypoints and RANSAC. All images are then warped to the Tele frame and cropped accordingly. When automatic registration fails, manual alignment is used. Despite this correction, residual geometric misalignment remains due to parallax and non-planar scene structure — motivating our use of alignment-aware HSR models. Finally, all images are downsampled to match the hyperspectral resolution, originally $194 \times 259$ cropped to $192 \times 256$ for convenience, and spectral grid is resampled to conventional 400–730 nm, $n = 34$, to optimize computational costs.

---

[1]The name *Doomer* is inspired by the subcultural aesthetic: most scenes were collected under gloomy or overcast weather conditions, in contrast to the brightly lit existing datasets.

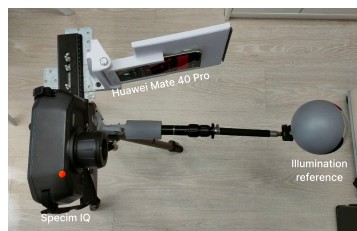 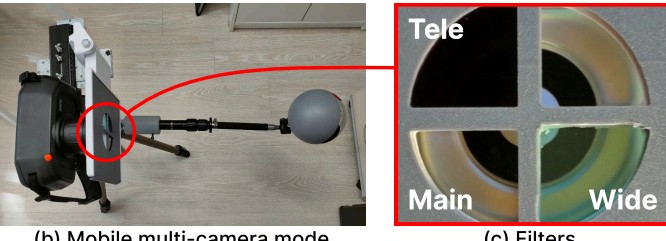

(a) Hyperspectral mode       (b) Mobile multi-camera mode       (c) Filters

Figure 4: *Capture setup for the Doomer dataset.* (a) Smartphone holder rotated to allow hyperspectral capture via Specim IQ. (b) Smartphone repositioned for simultaneous multi-camera RGB capture. (c) External spectral filters mounted on Tele and Wide cameras to induce spectral diversity.

## 5 METHOD

The task of MI-HSR involves predicting a hyperspectral image aligned to a target viewpoint (the Main camera), given RGB images from multiple spatially offset sensors:

$$\mathbf{I}_{\mathrm{Main}}, \mathbf{I}_{\mathrm{Tele}}, \mathbf{I}_{\mathrm{Wide}} \in \mathbb{R}^{h \times w \times 3}.$$

This setup introduces two key challenges: (i) the input views are misaligned due to differing camera geometries, and (ii) the available hyperspectral supervision $\mathbf{I}_{\mathrm{HS}} \in \mathbb{R}^{h \times w \times n}$ corresponds to a reference sensor not aligned with any RGB input. Our approach addresses both issues, one through pre-processing, the other through architectural design.

### 5.1 SUPERVISION WARPING VIA LEARNED OPTICAL FLOW

To leverage the hyperspectral reference image for training, we align it to the Main RGB view using learned optical flow (OF). Since spectral and RGB images differ in modality, we first compute a color projection $C \in \mathbb{R}^{n \times 3}$ to transform $\mathbf{I}_{\mathrm{HS}}$ into an RGB approximation:

$$C := \arg\min_{C} \|\mathbf{I}_{\mathrm{HS}} C - \mathbf{I}_{\mathrm{Main}}\|_2^2.$$

Given $\mathbf{I}_{\mathrm{HS}} C$ and $\mathbf{I}_{\mathrm{Main}}$, we estimate a dense correspondence field $\mathbf{D} \in \mathbb{R}^{h \times w \times 2}$ using a pre-trained OF model $\mathcal{F}$ (Sun et al., 2018):

$$\mathbf{D} := \mathcal{F}(\mathbf{I}_{\mathrm{HS}} C, \mathbf{I}_{\mathrm{Main}}).$$

This flow is used to warp the hyperspectral GT to the Main view:

$$\mathbf{I}_{\mathrm{HS}}^{\mathrm{w}} := \mathcal{W}(\mathbf{I}_{\mathrm{HS}}, \mathbf{D}), \quad M := \lfloor \mathcal{W}(J, \mathbf{D}) \rfloor,$$

where $M$ is a binary mask indicating valid visible pixels, $\mathcal{W}$ is warping operator, $J$ is matrix of ones. This enables aligned supervision for training and pointwise loss computation $\mathcal{L}_1(\mathbf{I}_{\mathrm{HS}}^{\mathrm{w}} \odot M, \hat{\mathbf{I}}_{\mathrm{HS}} \odot M)$

### 5.2 IMPLICIT CROSS-VIEW ALIGNMENT IN NETWORK DESIGN

Even with warped supervision, the three input views remain spatially misaligned. Direct calibration or flow-based alignment is possible but impractical in general-purpose settings. Instead, we encode alignment into the architecture itself, drawing inspiration from recent findings that transformer attention can perform implicit alignment across modest viewpoint shifts (Shi et al., 2022).

We adopt MSFN (Wu et al., 2024) as our base model, which consists of spatial and spectral transformer-based U-Nets. However, its original design applies spectral modeling before spatial, which is unsuitable for misaligned inputs as spectral attention assumes spatial coherence across input channels, which multi-camera setups violate.

### 5.3 PROPOSED ARCHITECTURE: MI-MSFN

We propose a revised architecture, Multi-Image MSFN (MI-MSFN), specifically tailored to address input misalignment and heterogeneous spatial content. Here, we reverse the order of modules,

applying spatial transformer blocks before spectral ones (Fig. 5). This allows the network to first establish spatial correspondences across the Main, Tele, and Wide inputs thus effectively aligning features by attention before fusing spectral information. We also remove the skip paths to enforce that all feature propagation passes through alignment-aware components.

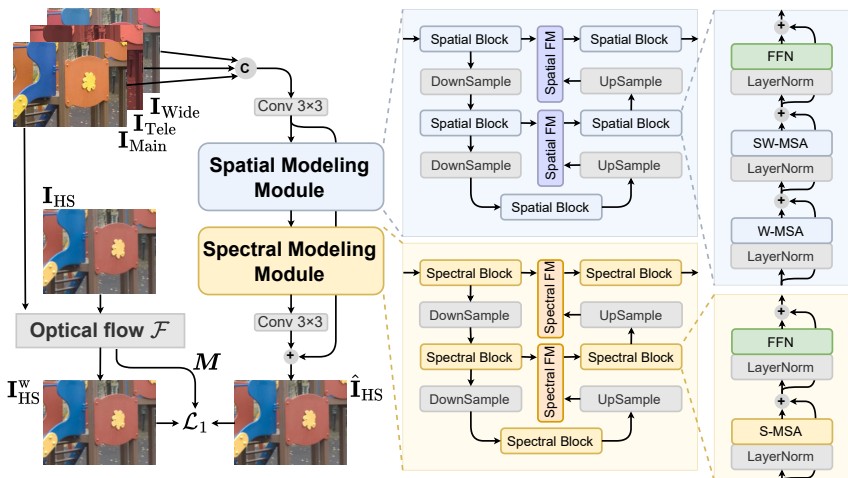

Figure 5: Overview of the proposed MI-MSFN architecture for multi-image hyperspectral reconstruction. It integrates implicit alignment across misaligned inputs via spatial-first attention, followed by spectral modeling. For detailed module definitions (*Spectral FM*, *Spatial FM*), refer to Sec. A.5

## 6 EXPERIMENTS

### 6.1 EXPERIMENTAL SETUP

**Dataset** All the experiments were conducted on our *Doomer* dataset as it is the only available one for MI-HSR task. Experiments on single-image HSR were only conducted for comparison to MI-HSR, thus featuring the same dataset.

**Metrics** We adopt well-known Peak Signal-to-Noise Ratio (PSNR) and Spectral Angle Mapper (SAM). We also introduce Normalized Spectral Error (NSE), which serves as an alternative to MRAE as it does not penalize having few dark bands and reflects the integral nature of radiance:

$$\mathrm{NSE}(\hat{\boldsymbol{r}}, \boldsymbol{r}) = \frac{||\hat{\boldsymbol{r}} - \boldsymbol{r}||_1}{||\mathbf{r}||_1} \cdot 100\%, \quad \boldsymbol{r}, \hat{\boldsymbol{r}} \in \mathbb{R}^n.$$

When comparing predictions to warped GT, some pixels were missing or invalid, so we masked out positions at $\mathbf{m}(x, y) = 0$.

**Implementation details** We split our dataset into *train* and *test* subset in proportion 4:1. In all the experiments we used Adam optimizer with learning rate of 0.0004 inherited from MSFN (except for AWAN at Sec. 6.3, which showcased the necessity of polynomial scheduler with power of 1.5 starting at learning rate of 0.0001). We trained all the networks on random $64 \times 64$ patches for $10^4$ epochs. Every 30 epochs we ran evaluation loop; the final model is the one with the best mean absolute error on the aligned test set. Each training procedure was re-run 10 times with different random seeds, yielding 10 distinct quantitative results. We summarize them by reporting mean and standard deviation.

### 6.2 RESULTS

We trained the proposed MI-MSFN network using two input configurations: single-image $\{\mathbf{I}_{\mathrm{Main}}\}$ and multi-image $\{\mathbf{I}_{\mathrm{Main}}, \mathbf{I}_{\mathrm{Tele}}, \mathbf{I}_{\mathrm{Wide}}\}$. To validate our supervision alignment (Sec. 5), following (Zhang et al., 2021), we report metrics both on aligned and original GT in Tab. 2. Our multi-image acquisition

system improves HSR by 3.56 dB PSNR, 38% SAM, 28% NSE. The lower standard deviation (1–3% of mean value) underlines the learning stability of the proposed MI-MSFN approach.

| Setting | Aligned GT | | | Original GT | | |
|---|---|---|---|---|---|---|
| | PSNR, dB ↑ | SAM, ° ↓ | NSE, % ↓ | PSNR, dB ↑ | SAM, ° ↓ | NSE, % ↓ |
| Single-camera | 26.30 ± 0.68 | 6.11 ± 0.40 | 12.71 ± 0.46 | 25.05 ± 0.48 | 6.25 ± 0.42 | 14.14 ± 0.67 |
| Multi-camera | **29.86 ± 0.21** | **3.77 ± 0.09** | **9.14 ± 0.20** | **27.88 ± 0.15** | **3.91 ± 0.08** | **10.35 ± 0.16** |

Table 2: Evaluations of MI-MSFN in single- and multi-camera settings. Mean and standard deviation of each metric is computed across 10 re-runs. The use of auxiliary cameras allows highly better HSR.

Fig. 6 presents a qualitative comparison between single-camera and multi-camera configurations. We show patch-level comparisons at a selected wavelength as well as radiance profiles at specific points of interest. The multi-camera system consistently recovers finer surface details that the single-camera setup fails to capture. In the first row, a printed symbol on a book cover is not registered by the multi-camera system at $\lambda = 700$ nm which complies with GT while single image HSR falsely reveals the symbol. In the second row, the single-camera system introduces spurious noise-like structures that do not correspond to any physical features. In the third row, the single-image setup fails to reconstruct radiometric intensity accurately, significantly misestimating the radiance of the paper sheet. We also showcase a failure case of our system in the fourth row. These results highlight the improved spatial and spectral fidelity enabled by multi-view fusion in the MI-HSR system.

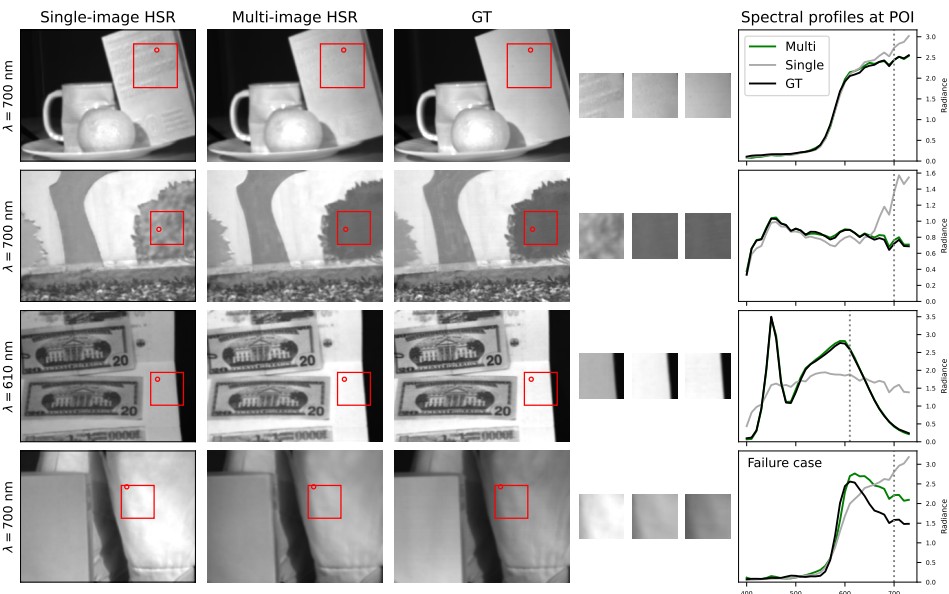

Figure 6: Qualitative comparison of MI-MSFN predictions under single-image and multi-image input settings. For each scene, a specific spectral band, region of interest, and point of interest are selected. Regions are enlarged and displayed side-by-side to highlight differences in spatial and spectral reconstruction quality. Note that the grayscale renderings correspond to individual spectral bands, as shown at the wavelength indicated on the left of each image.

## 6.3 ABLATION STUDY

We conducted two ablation studies, beginning with replacement of HSR network. We compared MI-MSFN against several approaches: HSCNN+ (Shi et al., 2018), AWAN (Arad et al., 2020; Li et al., 2020), MST++ (Cai et al., 2022) (winners of NTIRE 2018 (Arad et al., 2018), 2020 (Arad et al., 2020), 2022 (Arad et al., 2022) respectively) and MSFN (Wu et al., 2024) as shown in Tab. 3. Since we evaluate them in our misaligned multi-image setting they were not designed for, we also help them by warping inputs onto the Main camera view using the same pre-trained OF as before. In particular, Tab. 3 depicts the superiority of MI-MSFN over MSFN, justifying the proposed architectural changes.

| Method | OF | PSNR, dB ↑ | SAM, ° ↓ | NSE, % ↓ |
|---|---|---|---|---|
| HSCNN+ (Shi et al., 2018) | − | 26.28 ± 0.40 | 5.66 ± 0.13 | 12.20 ± 0.13 |
| | ✓ | 26.46 ± 0.64 | 5.76 ± 0.13 | 12.23 ± 0.17 |
| AWAN (Li et al., 2020) | − | 27.01 ± 0.79 | 4.56 ± 0.08 | 10.80 ± 0.10 |
| | ✓ | 26.41 ± 1.30 | 4.49 ± 0.17 | 10.68 ± 0.20 |
| MST++ (Cai et al., 2022) | − | 29.01 ± 0.25 | 4.18 ± 0.15 | 9.86 ± 0.17 |
| | ✓ | 29.37 ± 0.31 | 4.11 ± 0.07 | 9.66 ± 0.11 |
| MSFN (Wu et al., 2024) | − | 29.18 ± 0.21 | 3.91 ± 0.13 | 9.77 ± 0.13 |
| | ✓ | 29.59 ± 0.21 | 3.93 ± 0.13 | 9.33 ± 0.23 |
| MI-MSFN (ours) | − | **29.86 ± 0.21** | **3.77 ± 0.09** | **9.14 ± 0.20** |

Table 3: Ablation study on HSR model showing MI-MSFN being the best choice for MI-HSR.

| Active cameras | | | | | |
|---|---|---|---|---|---|
| Main | Tele | Wide | PSNR, dB ↑ | SAM, ° ↓ | NSE, % ↓ |
| ✓ | | | 26.30 ± 0.68 | 6.11 ± 0.40 | 12.71 ± 0.46 |
| ✓ | | ✓ | 26.34 ± 0.67 | 5.77 ± 0.31 | 12.41 ± 0.46 |
| ✓ | ✓ | | 29.63 ± 0.28 | 3.97 ± 0.11 | **9.08 ± 0.33** |
| ✓ | ✓ | ✓ | **29.86 ± 0.21** | **3.77 ± 0.09** | 9.14 ± 0.20 |

Table 4: Ablation study on cameras configuration. Each camera brings an improvement in SAM.

The second ablation study evaluates contribution of each auxiliary camera. It extends Tab. 2 by two more settings: $\{I_{Main}, I_{Tele}\}$ and $\{I_{Main}, I_{Wide}\}$. Results are given in Tab. 4. We can see that just by adding the Tele-camera we are already close to the performance of all three. Notably, this camera was occluded by the red filter (Fig. 2) and most examples from qualitative comparison (Fig. 6) showcase how our setup handles discrepancy in red spectral range. In terms of NSE-metric, the combination of Main and Tele even outperforms everything, but statistically insignificantly.

## 6.4 EFFECT OF NOISE IN HYPERSPECTRAL CAMERA

Hyperspectral image registration is a process of registering photons, which means our GTs are inherently noisy just like consumer CMOS sensors. Since noise is unpredictable, HSR quality has a physically-reasoned insurmountable boundary. To quantify this boundary, we captured series of HSIs of static scenes with color charts under two different light conditions. The metrics were calculated between a fixed image from a series and the average of remain images, see Tab. 5. The estimated values should be interpreted as approximate bound for any MI-MSR network trained and tested on Doomer dataset. Comparable metric values may indicate overfitting or sensitivity to sensor noise.

| Light conditions | PSNR, dB ↑ | SAM, ° ↓ | NSE, % ↓ |
|---|---|---|---|
| Bright scenario (outdoor sunny) | 41.32 ± 3.04 | 0.71 ± 0.11 | 3.13 ± 0.55 |
| Medium brightness (indoor LED) | 37.81 ± 3.49 | 1.02 ± 0.12 | 4.46 ± 1.27 |

Table 5: Metrics measured between several Specim IQ shots of the same scene. These values suggest rough estimates of the highest probable quality of HSR on Doomer.

## 7 LIMITATIONS

The trained models in this work are tightly coupled to the spectral characteristics and quantum efficiency of the specific RGB sensors and optical filters used in our acquisition setup. As a result, deploying the system with different hardware configurations would require collecting a new dataset and retraining the model. This limits immediate out-of-the-box generalization. Future research could explore strategies such as sensor-specific domain adaptation, transfer learning, or SSF-invariant reconstruction frameworks to improve reproducibility and portability across different camera systems.

## 8 CONCLUSION

This work rethinks HSR through the lens of practical acquisition. By moving beyond single RGB image constraints, we demonstrate that leveraging multiple smartphone cameras with carefully chosen spectral filters can significantly enrich the input signal and reduce ambiguity in reconstruction. Our proposed MI-HSR framework shows that spectral diversity even when captured with commodity hardware can close the gap between simulated and real-world hyperspectral imaging. The introduction of the Doomer dataset marks an important step forward for benchmarking in this space, enabling systematic evaluation of multi-view HSR under realistic conditions. Empirical results validate both our hardware configuration and model design, suggesting a promising direction for low-cost, deployable hyperspectral imaging systems. Looking ahead, we aim to further explore the temporal dimension for dynamic scenes, optimize for energy-efficient mobile deployment, and investigate more principled learning paradigms under limited supervision or device mismatch.

## 9 REPRODUCIBILITY STATEMENT

We provide the code of the proposed MI-MSFN model along with the Doomer dataset under a link given in Sec. A.1. The proposed acquisition setup and the data preprocessing steps are described in Sec. 4.

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

# A    TECHNICAL APPENDICES AND SUPPLEMENTARY MATERIAL

To ensure disambiguation, here is the list of some designations used throughout the main paper and this document:

- Photometric normalization — division of a camera sensor signal by ISO (if applicable) and exposure time. Black current subtraction is also applied beforehand.
- **c** — a photometrically normalized signal (probably noisy) from smartphone camera(s). Can be $\in \mathbb{R}^3$ or $\in \mathbb{R}^{3k}$ depending on context.
- $\bar{\mathbf{c}}$ — a photometrically normalized signal as it would be in an ideal world without noise.
- **r** — a photometrically normalized radiance spectrum. Is assumed to be noiseless (however, in Sec. 6.4 of the main paper we discuss the outcomes of this assumption).

## A.1    CODE AND DATA

Click here to download.

## A.2    ESTIMATION OF SPECTRAL SENSITIVITY FUNCTIONS AND TRANSMITTANCE FUNCTIONS

**Spectral sensitivity functions.** To estimate SSF of a smartphone camera, we acquired 25 sample pairs of $c_i \in \mathbb{R}^3$ and $r_i \in \mathbb{R}^n$ corresponding to flat-field illumination (FFI) of narrow-band LED light sources in an integrating sphere. For each LED $i$, we took a photo of FFI (Fig. 7), extracted the central 100×100 patch from it, photometrically normalized it and computed the channel-wise average to get $c_i$. Then we measured the LED radiance spectrum $r_i$ using an X-Rite i1 Pro spectrophotometer with the help of spotread routine from Argyll color management system.

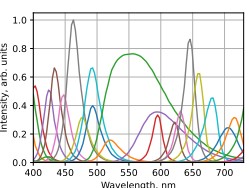


(a) LEDs' radiance spectra          (b) Sample LED FFI photo

Figure 7: Training data for SSF estimation: (a) spectra of 25 LEDs measured using X-Rite i1 Pro; (b) an example of raw FFI photo. Notice the vignetting effect. To mitigate this effect, channel-wise average of a tiny central patch (in red) was used as a sample $c_i$

Given the measurements $\{c_i, r_i\}_{i=1}^{25}$, the spectral sensitivities estimation problem can be formulated as a regularized quadratic optimization problem:

$$\min_{\boldsymbol{S}} \quad \sum_{i=1}^{25} \|c_i - \boldsymbol{S}^\top r_i\|_2^2 + \lambda \|\boldsymbol{D}\boldsymbol{S}\|_2^2 \tag{1}$$
$$\text{s. t.} \quad \boldsymbol{S} \geq 0$$

where the regularization term $\lambda \|\boldsymbol{D}\boldsymbol{S}\|_2^2$ imposes smoothness, $\boldsymbol{D}$ is the second-order derivative operator. The objective (1) was minimized using the Adam optimizer.

To validate the estimated $\hat{\boldsymbol{S}}$, we captured a color rendition chart by the Specim IQ hyperspectral camera and all three smartphone cameras. The HSI was calibrated (more on that in the next paragraph) and projected onto each camera's sensor space to yield RGB predictions. We compared the predicted and actual RGBs of the color patches: pixels of a patch were pixel-wise averaged (Fig. 12 gives a sight on how such patches look like). The average angular error of color reproduction was $\approx 1°$ — indistinguishable to a human eye.

**Specim IQ calibration.** Spectral measurements of X-Rite i1 Pro and Specim IQ are actually inconsistent. It is not suprising: Specim IQ is not designed to be photometrically accurate. Instead, it

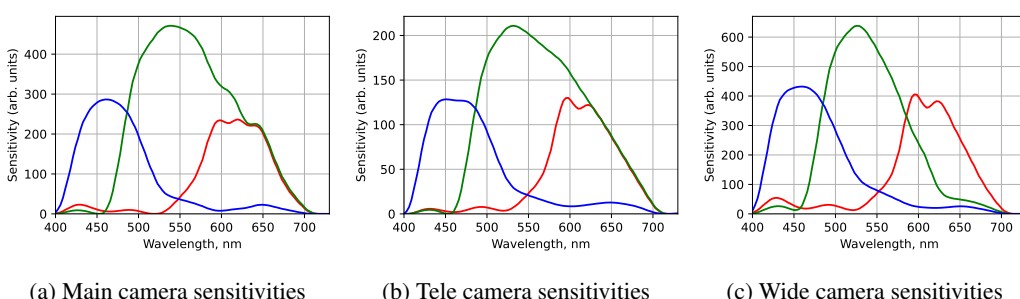

(a) Main camera sensitivities     (b) Tele camera sensitivities     (c) Wide camera sensitivities

Figure 8: Estimated smartphone cameras spectral sensitivities.

allows to estimate band-wise relations between the *calibration target* in a scene and other objects. However, if we estimate the band-wise calibration divisor between X-Rite i1 Pro and Specim IQ, we can futher use it to obtain physically-correct HSIs. We illuminated the integration sphere using a mixture of LEDs that yields even spectral power distribution, obtained X-Rite and Specim IQ measurements of it and divided one by another:

$$\boldsymbol{k} = \boldsymbol{r}_{\mathrm{specim}} \oslash \boldsymbol{r}_{\mathrm{xrite}},$$

The divisor is applied to every HSI filmed by the Specim IQ in this research.

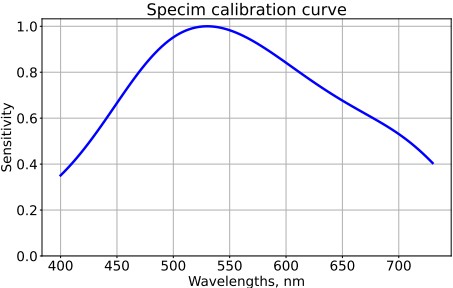

Figure 9: The estimated Specim IQ calibration divisor

Since the operational range of Specim IQ is 400–1000 nm and the one of X-Rite is 380–730 nm, all our spectral measurements are defined on their intersection: 400–730 nm.

**Transmittance functions of filters** were measured using an SF-2000 spectrophotometer.

A.3   SPECTRAL UNCERTAINTY

The inspiration comes from the well-known conditional entropy $H\left(\xi \mid \eta\right)$, which quantifies the information loss of a latent random variable $\xi$ when observing the outcomes of another random variable $\eta$:

$$H\left(\xi \mid \eta\right) = \mathbb{E}_{x \sim \eta} H\left(\xi \mid \eta = x\right), \tag{2}$$

where $H\left(\xi \mid \eta = x\right)$ is the ordinary entropy of $\xi$ given event $\eta = x$. However, entropy is designed especially for discrete-distributed variables of categorical type. When uncertainty of a categorical random variable is indeed best described by the entropy of its distribution, for continuous random variables variance is a better choice. When dealing with random vectors, variance becomes a matrix, so we consider the trace of it as a reasonable summary. If we put the trace of variance instead of the entropy in (2) and substitute $\xi = \mathbf{r}, \eta = \mathbf{c}$, we get the spectral uncertainty of our filters-modified optical system:

$$v(\boldsymbol{F}) = \mathbb{E}_{\mathbf{c}}\left[\operatorname{tr} \operatorname{Var}_{\mathbf{r}}(\mathbf{r} \mid \mathbf{c})\right],$$

where the dependence on $\boldsymbol{F}$ is hidden inside the relationship between $\mathbf{r}$ and $\mathbf{c}$. However, $\mathbf{r}$ and $\mathbf{c}$ are not random variables until we define them so. Let $\mathbf{r}$ have a discrete distribution $p\left(\mathbf{r}_i\right)$ over a finite set

of radiance spectra $\mathcal{R} = \{\mathbf{r}_1, ..., \mathbf{r}_N\}$ derived from a dataset of HSIs. Such definition reflects the *a priori* information about natural radiance. In the relation $\mathbf{c} = \boldsymbol{S_F}\mathbf{r} + \mathbf{n}$ only noise $\mathbf{n}$ is yet to define. Let $\bar{\mathbf{c}} = \boldsymbol{S_F}\mathbf{r}$ be an unnoised camera response. We model noise as

$$\mathbf{n} \mid \bar{\mathbf{c}} \sim \mathcal{N}(\mathbf{0}, \boldsymbol{\Sigma}(\bar{\mathbf{c}})), \tag{3}$$

$$\boldsymbol{\Sigma}(\bar{\mathbf{c}}) = \mathrm{diag}[\sigma_1^2(\bar{c}_1); ...; \sigma_{3k}^2(\bar{c}_{3k})], \tag{4}$$

where $\sigma_i(\cdot)$ is expanded in (5) later in the chapter.

Now, when we defined $p(\mathbf{c} \mid \mathbf{r})$, we can derive from the Bayesian rule:

$$p(\mathbf{r}_i \mid \mathbf{c}) = \frac{p(\mathbf{c} \mid \mathbf{r}_i)\, p(\mathbf{r}_i)}{\sum_{i=1}^N p(\mathbf{c} \mid \mathbf{r}_i)\, p(\mathbf{r}_i)},$$

and further:

$$\mathbb{E}_{\mathbf{r}}(\mathbf{r} \mid \mathbf{c}) = \sum_{i=1}^N p(\mathbf{r}_i \mid \mathbf{c})\, \mathbf{r}_i,$$

$$\mathrm{tr}\, \mathrm{Var}_{\mathbf{r}}(\mathbf{r} \mid \mathbf{c}) = \sum_{i=1}^N p(\mathbf{r}_i \mid \mathbf{c})\, \|\mathbf{r}_i - \mathbb{E}_{\mathbf{r}}(\mathbf{r} \mid \mathbf{c})\|_2^2.$$

Now we have an expression depending on $\mathbf{c}$ which we should take $\mathbb{E}_{\mathbf{c}}$ of. However, this is intractable. So we should resort to Monte Carlo method by sampling $\mathbf{c}$ according to its definition: first take random $\mathbf{r}^* \in \mathcal{R}$, then sample $\mathbf{c}$ from $\mathcal{N}(\boldsymbol{S_F}\mathbf{r}^*, \Sigma(\boldsymbol{S_F}\mathbf{r}^*))$. In our experiments, $2^{20}$ samples were sufficient to achieve 0.5% of relative standard deviation.

To derive $\mathcal{R}$, we first extracted every 29×29th pixel of the KAUST dataset (Li et al., 2021), which resulted in 559,921 samples. However, this is a dataset of reflectances, so we converted them to radiance spectra via multiplying it by a spectrum of gray ball under overcast weather condition. This way, we assured that $\mathcal{R}$ mostly fairly represents our future *Doomer* dataset. To reduce computational requirements, we also compressed $\mathcal{R}$ to the size of $N = 1024$ by running K-Means algorithm and employing clusters' centers. We assigned $p(\mathbf{r}_i)$ to be the share of initial 559,921 radiance spectra that belong to the cluster around $\mathbf{r}_i$.

Unfortunately, our initial implementation of spectral uncertainty contained a bug in the code, so the selected filters pair (Fig. 2) is not actually the best according to the described criterion. The bug has been spot long after the dataset was collected. In Fig. 10 we show the best filters selected by the criterion after the bug fix. We also probe those filters in simulated settings in Sec. A.6

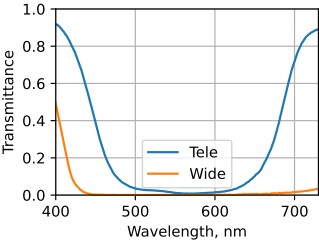

Figure 10: Best filters after the bug fix

**Experimental justification.** Having the estimated spectral characteristics of our filtered smartphone cameras setup ($\boldsymbol{S_F}$) and the sensor noise model (5), we find ourselves in a position where we can simulate images of the triple-camera setup from HSIs and run experiments on simulated data. Therefore, we can support the proposed spectral uncertainty criterion by simulating input images $\mathsf{I}_{\mathrm{Main}}, \mathsf{I}_{\mathrm{Tele}}, \mathsf{I}_{\mathrm{Wide}}$ given filters, measuring HSR performance and plotting it against $v(\boldsymbol{S_F})$.

We chose 12 random filter pairs and simulated 12 versions of our dataset. For each version, MI-MSFN was trained and evaluated. Fig. 11 shows the dependence of three performance metrics on the proposed spectral uncertainty criterion. A strong correlation is observed, as the Pearson correlation coefficient has an absolute value greater than 0.8.

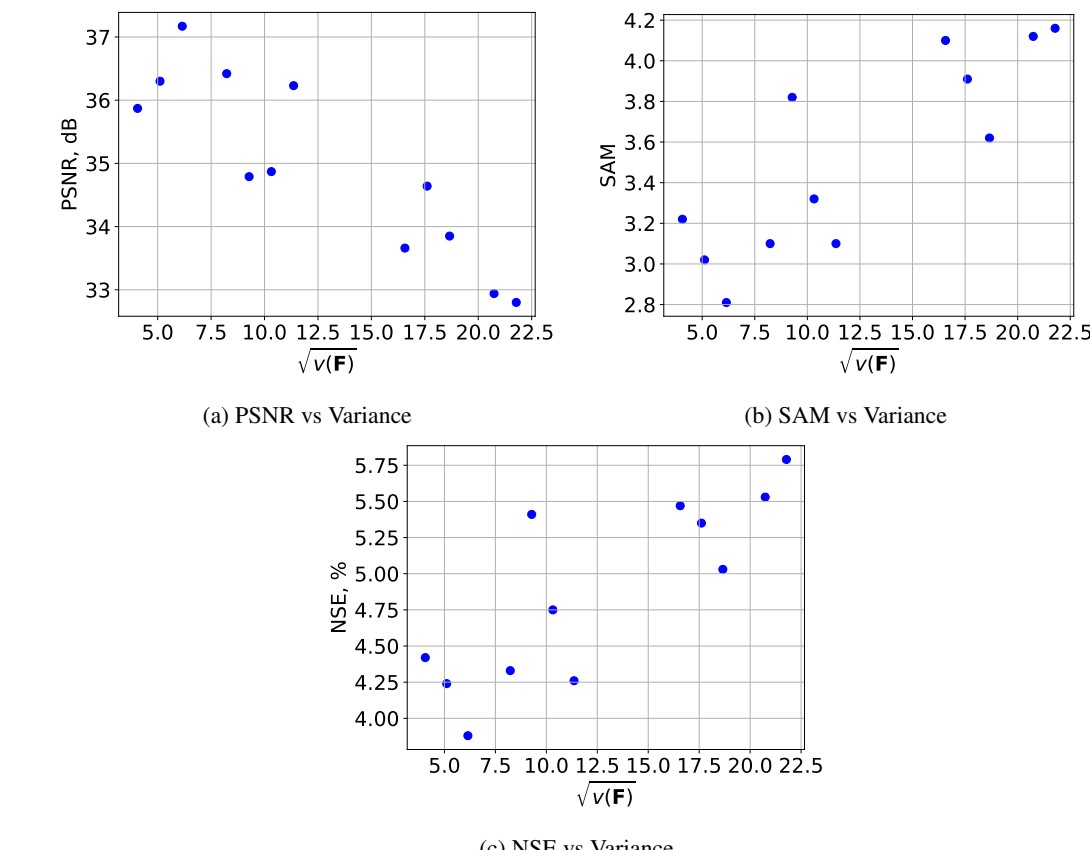

(a) PSNR vs Variance

(b) SAM vs Variance

(c) NSE vs Variance

Figure 11: Relation between PSNR, SAM, NSE in simulated settings and the proposed spectral uncertainty criterion.

**Noise model.** Let $\bar{c}_i$ be a photometrically-normalized ground-truth signal level at channel $i$, $t$ be exposure time, $g$ be ISO. The expected collected charge at image sensor is $\bar{c}_i t$. Following Foi et al. (2008), we model noise of the collected charge as $\varepsilon \sim \mathcal{N}(0, \alpha_i \bar{c}_i t + \beta_i)$, where $\alpha_i$, $\beta_i$ are parameters of the model. The final value in the raw image is further amplified by $g$ along with the noise: $(\bar{c}_i t + \varepsilon)g$. Foi et al. also introduce additional noise after amplification, but we neglect it for simplicity. The observed photometrically-normalized signal is then given by:

$$c_i = \frac{(\bar{c}_i t + \varepsilon)g}{tg} = \bar{c}_i + \frac{\varepsilon}{t}.$$

To be consistent with (3) and (4), we define $\sigma_i(x)$ as:

$$\sigma_i(\bar{c}_i) = \frac{\sqrt{\alpha_i \bar{c}_i t + \beta_i}}{t} \tag{5}$$

We estimate parameters $\alpha_i$, $\beta_i$ of this model by plotting $\sigma_i t$ against $\bar{c}_i t$ in Fig. 12.

The only problem left in (5) is the hidden dependence on $t$. Given simulated $\bar{\mathbf{c}} = \boldsymbol{S_F}\mathbf{r}$, we cannot add noise properly unless we know exposure time $t$. In real-life setting, exposure time is determined by a camera automatically depending on the brightness of a scene. For simplicity, we model this in a piecewise-power fashion. We gathered 1000 raw images from phone's gallery and plotted $t$ against the average of photometrically normalized signals across all channels, height and width in an image $c = \frac{1}{h \times w \times 3} \sum_{i,j,k} \mathbf{l}_{i,j,k}$ (Fig. 13). The fitted function allows to estimate $t$ given $\bar{c}_i$.

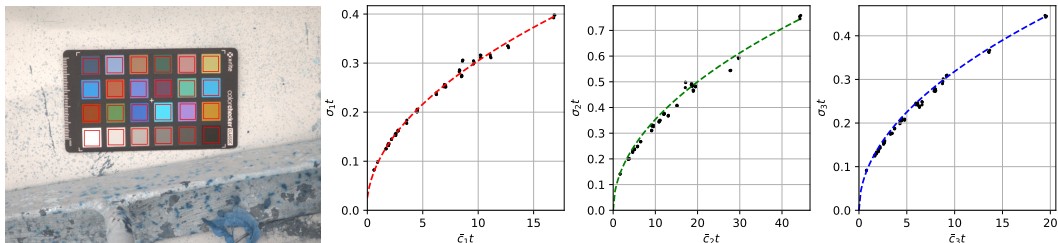

Figure 12: Plotting $\sigma_i t$ against $\bar{c}_i t$ for the Telephoto camera based on a single shot. Each data point (24 in total) is given by channel-wise mean and std dev of a color patch (left, in red). In each plot, point cloud is approximated by a dashed line (5) using least-squares; each color channel $i$ gets its own noise parameters $\alpha_i$, $\beta_i$ (right).

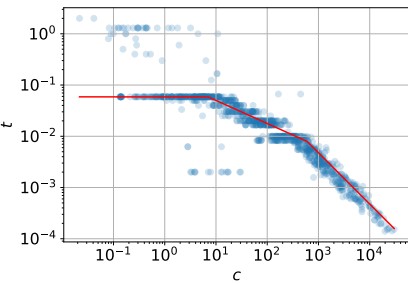

Figure 13: How exposure time depends on scene brightness. The piecewise-power function plotted in red line is used to model the exposure time given the average of a photometrically normalized image.

### A.4    More on Doomer Dataset

In the full public release of Doomer dataset we will provide 3 main versions:

1. **RAW version**. Images from all cameras before any preprocessing.

2. **Single-camera version**. Pairs of a Main-camera photo and an HSI. The images are preprocessed and share the same FoV and spatial resolution.

3. **Multi-camera version**. Quadriples of Main-, Tele- and Wide-camera photos and HSI. The images are preprocessed and share the same FoV and spatial resolution.

Both the single- and multi-camera versions will also have two subversions depending on what spectral resolution of an HSI a user needs. Our resources allowed us to capture spectral radiance with the sampling interval of 3 nm. However, most of hyperspectral images have that of 10 nm as a compromise between precision and computational efficiency. A short summary of each version provided in Tab. 6. At the time of submission, only *emphasized* items are made available.

The preprocessing pipeline of RAW images consists of these steps: demosaicing, black current subtraction, flat field calibration and photometric normalization. For hyperspectral images, preprocessing pipeline includes: black current subtraction, flat field calibration, radial distortion correction and photometric normalization. The last step involves only division by the exposure time (Specim IQ has no ISO setting) and band-wise division by the calibration divisor **k**.

For all HSIs, we will provide a manually annotated binary mask that specifies the location of the gray ball. The reflectance spectrum of the gray ball, measured with the X-Rite spectrophotometer, will be included in the dataset. Also, we will expand it by adding more scenes and annotate each scene with tags depending on its contents.

Unlike previously published datasets, Doomer contains multiple real RGB images along with ground-truth HSI and illumination reference. This combination enables the exploration of various computational photography problems, such as:

1. *White point estimation* both in spectral and RGB forms.

| Dataset version | Smartphone images | HSIs spectral range (step) | | FoV matching |
|---|---|---|---|---|
| RAW | Main, Tele, Wide | Original 400–1000 nm (3 nm) | | – |
| 1-camera | Preprocessed Main camera image | Preprocessed 400–730 nm (10 nm) | Preprocessed 400–730 nm (3 nm) | ✓ |
| *3-camera* | *Preprocessed and concatenated Main, Tele, Wide images* | *Preprocessed 400–730 nm (10 nm)* | Preprocessed 400–730 nm (3 nm) | ✓ |

Table 6: Doomer Dataset versions

2. *Illumination distribution estimation.* The need to estimate a distribution of illumination in a scene arises from the complexity of natural scenes. In such environments, a single global white point may be insufficient for accurate image processing and color correction (Ershov et al., 2023).

3. *Color space transform.* Since there is a known white-point for each scene, it is possible to do precise chromatic adaptation and color signals for different cameras.

4. *Hyperspectral Reconstruction.* Reconstruction from single or multiple cameras. Reconstruction of different types of spectra — radiance or reflectance.

The data preprocessing pipeline was described in Sec. 4 and is briefly illustrated in Fig. 14.

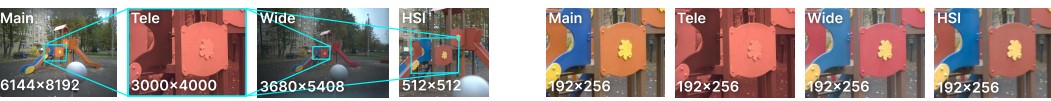

Figure 14: *Spatial preprocessing pipeline.* Left: geometric alignment of RGB views using SIFT + RANSAC for consistent cross-camera registration. Right: field-of-view normalization and resolution matching across RGB and hyperspectral modalities.

## A.5 REST OF THE MI-MSFN ARCHITECTURE

Since the proposed neural network is rather a minor contribution and differs from MSFN only by the arrangement of blocks, we only gave a broad view on it in the main part of the paper. Some blocks were left unexpanded, which we fix by showing the missing details here in Fig. 15.

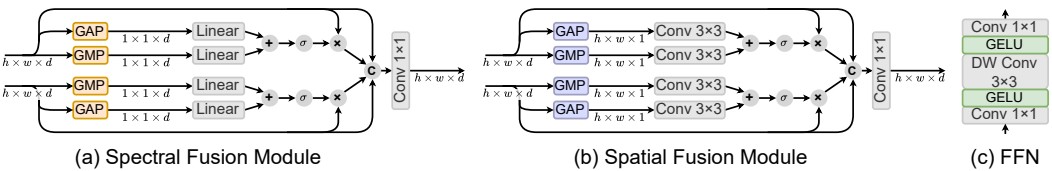

Figure 15: Illustration of fusion modules and feed-forward network in MI-MSFN. GAP and GMP stand for global average and max pooling respectively.

The remaining disambiguation consists of abbreviations expansion: S-MSA = spectral multihead self-attention (Cai et al., 2022); SW-MSA = shifted window multihead self-attention (Liu et al., 2021); W-MSA = window multihead self-attention (Dosovitskiy et al., 2020).

## A.6 ADDITIONAL EXPERIMENTS ON SIMULATED DATA

Unlike the real-life conditions, the simulated images are perfectly aligned to HSI and between each other. Evaluation on a simulated dataset then gives us a sight on how big the performance increase

caused by the triple-camera setup could be if there was no misalignment at all. The results are shown in Tab. 7

| Setting | PSNR, dB ↑ | SAM, ° ↓ | NSE, % ↓ |
|---|---|---|---|
| Single-camera | 28.79 ± 0.46 | 5.40 ± 0.22 | 8.46 ± 0.37 |
| Multi-camera | **36.84 ± 0.39** | **2.69 ± 0.02** | **3.88 ± 0.06** |
| Multi-camera (filters from Fig. 10) | **37.90 ± 0.31** | **2.33 ± 0.06** | **3.43 ± 0.08** |

Table 7: Evaluations of MI-MSFN in single- and multi-camera settings on simulated data. Evaluation of the better filters combination is provided separately

The triple-camera setup in simulated conditions yields $\sim 50\%$ improvement in terms of SAM and NSE and +8 dB PSNR. Recalling the $\sim 30\%$ improvement in the real-life conditions, we can conclude that our results in the main part of the paper are reasonable.

Tab. 7 also includes results for the filters selected after we fixed the bug in our code (Fig. 10). It suggests that if we used those filters for the dataset collection, we could have achieved even better results on the real data in the main paper. Also, this simulation result strengthens the reasonableness of spectral uncertainty criterion.

## A.7 COMPUTATIONAL RESOURCES

All the network trainings (Tab. 2, 3, 4) were executed on a cluster, allowing us to run up to 20 jobs in parallel. This was especially handy because we needed to run each experiment 10 times with different random seeds to ensure statistic significance. The cluster randomly assigned jobs to nodes, each having multiple GPUs. However, any job only used one GPU at a time. A GPU assigned to a job could be one of the following: NVIDIA L40, L40s and H100. Every training consumed < 23 GB of GPU memory and lasted no longer than 8 hours. 5 CPU cores were occupied by data loader; the total CPU memory consumption was < 8 GB.

