# OpenReview forum: "See the World through Color-tinted Glasses for Better Hyperspectral Reconstruction"
_ICLR.cc/2026/Conference — ICLR 2026 Conference Withdrawn Submission_

### Official Review · Reviewer_qJXA · 2025-10-26

**Soundness:** 3
**Presentation:** 3
**Contribution:** 3
**Rating:** 6
**Confidence:** 5

**Summary:**

The paper proposes a practical multi-image-to-hyperspectral reconstruction (MI-HSR) system that enhances spectral fidelity by augmenting a smartphone’s auxiliary cameras with spectral filters. It introduces a new dataset (Doomer), an information-theoretic filter selection strategy, and a transformer-based reconstruction model (MI-MSFN) that handles misalignment and heterogeneity in input views.

**Strengths:**

* Leverages triple-camera smartphones with external filters to capture spectrally diverse data without modifying internal hardware.

* The Doomer dataset is a significant contribution in the right direction. The dataset captures real-world, misaligned, multi-camera RGB and HSI captures under uncontrolled lighting with colour references.

* Filter choice is optimised using a spectral uncertainty minimisation strategy grounded in information theory.

* The MI-MSFN architecture introduces spatial-first transformer modelling, outperforming prior art in multi-camera fusion.

* Ablation studies, noise model analysis, and comparison with state-of-the-art (SOTA) HSR methods are thorough and statistically sound.

**Weaknesses:**

* The system is tightly coupled to specific hardware (Huawei Mate 40 Pro + selected filters), limiting generalizability. Although acknowledged in limitations, the lack of generalisation experiments (e.g., across different phones or filter sets) weakens real-world applicability.

* The authors admit a bug in the original filter selection process after dataset collection (Appendix A.3). While they justify the selection via later simulations, this undermines claims of optimality in filter choice.

* There’s insufficient discussion comparing the capture latency, energy consumption, or computational overhead of the proposed system vs. prior low-cost or MSFA-based systems.

* The study could be significantly strengthened by experiments showing whether MI-MSFN generalises to synthetic RGBs or different camera systems, or by including sensor-invariant training strategies.

*  “...we futher use it to obtain physically-correct HSIs” in it further is misspelled.

**Questions:**

* Could the authors clarify how confident they are that the chosen filters (despite the bug) still perform near-optimally in real-world data capture? Can they provide performance comparisons between the originally used filters and the corrected optimal ones?

* How dependent is the method on exact smartphone camera specifications (e.g., SSF, baseline distance, lens properties)? Is domain adaptation or fine-tuning feasible for other devices?

* Can you provide inference latency and memory use for MI-MSFN? Could it realistically run on a mobile device or edge platform?

* Would it be possible to include simulated data from other devices during training to improve generalizability?

* Since the authors mention temporal HSR in future work, do they anticipate further gains by incorporating video input or temporally adjacent views?

---

### Official Review · Reviewer_Lx7D · 2025-10-27

**Soundness:** 3
**Presentation:** 3
**Contribution:** 2
**Rating:** 2
**Confidence:** 3

**Summary:**

Hyperspectral reconstruction (HSR) from a single RGB image is a fundamentally ill-posed problem.
This work introduces a hyperspectral reconstruction framework that leverages multi-RAW inputs captured from a triple-camera setup (Main, Tele, and Wide cameras). By equipping the Tele and Wide cameras with carefully selected custom spectral filters and utilizing the multi-RAW data from all three cameras, the authors construct the Doomer dataset.

To address the spatial misalignment between each RAW image and the hyperspectral ground truth, the method first applies optical flow to align the hyperspectral reference to the Main camera view. Subsequently, the remaining spatial misalignments are corrected using the proposed MI-MSFN architecture, which revises the MSFN architecture. Overall, this work presents a promising direction toward low-cost and deployable hyperspectral imaging systems.

**Strengths:**

1) Novel multi-RAW hyperspectral reconstruction framework

The proposed hyperspectral reconstruction framework that leverages multi-RAW inputs captured from a triple-camera setup is novel and presents a fresh perspective on practical hyperspectral imaging.

2) Useful architecture for multi-camera systems

The proposed architecture effectively addresses the misalignment and heterogeneous spatial content issues inherent in multi-image inputs. This design is not only valuable for hyperspectral reconstruction but can also benefit other imaging systems employing multiple cameras.

3) Open-source dataset contribution

Open-sourcing the hyperspectral reconstruction dataset designed for a triple-camera setup would be highly beneficial to researchers working on related topics, facilitating further research and development in this area.

**Weaknesses:**

1. Supervision of color projection $C$

In Section 5.1, it is unclear how the color projection module $C$ can be effectively supervised, given that the input hyperspectral image $I_{HS}$ and the target RGB image $I_{Main}$ are spatially misaligned. Since the optical flow–based warping appears to be applied after the training of the color projection $C$, it raises concerns about whether $C$ can be reliably learned under such spatial misalignment. A clarification or additional analysis on this point would strengthen the technical soundness of the method.

2. Limited novelty of the proposed architecture

Based on Sections 5.2 and 5.3, the main contribution appears to lie in reordering modules within MSFN and extending the model to handle multiple input images. While the implementation is well executed, the architectural novelty seems incremental relative to prior work, and additional insights into the design motivation or ablation studies could help emphasize its originality.

3. Limited relevance to the expected ICLR audience

Despite the novelty of the task and the well-constructed dataset, most of the ICLR audience would likely expect more substantial innovations in model architecture or training framework design. From this perspective, the paper’s contribution appears somewhat limited in methodological novelty. The work may be better suited to domain-specific venues such as IEEE Transactions on Image Processing (TIP) or IEEE Transactions on Computational Imaging (TCI), where hyperspectral imaging and multi-camera system design are central research themes. I respectfully recommend considering submission to one of these more relevant peer-reviewed journals.

**Questions:**

1) Selection of spectral filters

In Section 3.2, the authors carefully choose two auxiliary filters from 65 candidates. How sensitive is the method to this choice? If the deployed system uses a different, does performance degrade substantially?

2) Training stability under spatial misalignment

Following up on Weakness 1, was there any observed difficulty in training the color projection module $C$ due to spatial misalignment between the hyperspectral input and the RGB target images?

---

### Official Review · Reviewer_a8bG · 2025-10-31

**Soundness:** 3
**Presentation:** 3
**Contribution:** 3
**Rating:** 6
**Confidence:** 5

**Summary:**

The paper makes three primary contributions:

(1) Smartphone-based multi-channel imaging system:

The authors propose a low-cost acquisition system that attaches two custom spectral filters to the auxiliary cameras of a consumer smartphone, effectively transforming it into a 9-channel imaging device. The filter selection is optimized using information-theoretic criteria to minimize spectral uncertainty. This configuration is novel in the hyperspectral reconstruction (HSR) literature and shows significant improvements over RGB-only and naïve multi-view baselines.

(2) The Doomer dataset:

The authors introduce Doomer, the first benchmark dataset for multi-image hyperspectral reconstruction (MI-HSR). Each scene contains four captures—three from the smartphone’s cameras (two filtered and one unfiltered) and one from a hyperspectral reference camera—providing real-world, misaligned, multi-view data for training and evaluation.


(3) Reformulated transformer-based architecture:

The paper presents a principled redesign of transformer-based HSR models, introducing a spatial-first attention mechanism that performs implicit alignment across misaligned camera viewpoints. This design enables effective fusion of heterogeneous spatial inputs and substantially improves reconstruction quality and robustness.

**Strengths:**

(1) Novel and practical idea. The paper proposes a new paradigm for hyperspectral reconstruction (HSR) by leveraging the multi-camera system of consumer smartphones enhanced with two custom spectral filters. This idea is both innovative and practically relevant, as it enables real-time, multi-channel spectral capture without modifying internal hardware—a major step toward scalable, low-cost hyperspectral imaging. The filter selection is guided by information-theoretic spectral uncertainty minimization, giving the system a strong theoretical foundation

(2) Strong and Consistent Performance. The proposed MI-MSFN model achieves clear quantitative gains over single-camera and existing HSR baselines. Results are consistent across 10 random seeds, showing low variance and high stability, which supports the robustness of the method. Qualitative comparisons (e.g., Fig. 6) convincingly show better spectral fidelity and fewer artifacts in reconstructed images.

(3) Comprehensive Experimental Validation. The experiments are well-structured and thorough, covering both aligned and unaligned supervision, multiple baseline comparisons (HSCNN+, AWAN, MST++, MSFN), and real-world data. The authors provide detailed ablation studies analyzing: The contribution of each auxiliary camera (Tab. 4). The effect of optical flow alignment and model variants (Tab. 3). The inherent noise limits of the hyperspectral ground truth (Tab. 5). These studies demonstrate careful empirical reasoning and ensure that each design choice is justified.

(4) High-quality Dataset Contribution. The Doomer dataset is the first benchmark for multi-image hyperspectral reconstruction, featuring real, misaligned, multi-view smartphone captures with spectral references. The dataset is diverse (indoor/outdoor, varied lighting, gray reference ball) and fills a crucial gap between synthetic benchmarks and real-world use cases.

(5) Very good reproducibility. Code, pre-trained models, and even the dataset are submitted. The reproducibility is pretty good and other researchers can do further research based on the authors’ work.

(6) The paper is clearly written and logically structured, making complex technical ideas easy to follow. Each section—system design, dataset, method, and experiments—is cohesively connected, and figures (e.g., setup illustrations, architecture diagrams) effectively complement the explanations. The motivation, methodology, and results are communicated in a reader-friendly and professional manner, suitable for a ICLR paper.

**Weaknesses:**

(1) Limited Generalization Beyond the Specific Hardware Setup. The proposed system and dataset are tightly coupled to a single smartphone model (Huawei Mate 40 Pro) and its specific camera spectral sensitivities and custom filters. As acknowledged in the paper’s Limitations section, deploying this approach on other smartphones or camera systems would require collecting new datasets and retraining the model from scratch. This significantly limits the out-of-the-box generalizability and reproducibility of the method, reducing its immediate applicability across diverse hardware.

(2) No Evaluation of Cross-device or Real-world Deployment. While the system is marketed as “practical” and “mobile-friendly,” all experiments are conducted offline on a fixed dataset rather than real-time or in-device inference. There is no analysis of latency, power efficiency, or on-device feasibility, which would strengthen the argument for mobile deployment. The absence of cross-device testing makes it unclear how well the trained model performs on unseen camera hardware or under uncontrolled real-world conditions.

(3) Limited Comparison to Multi-camera or Hybrid Systems. The paper compares mostly to single-image HSR baselines, but not to prior multi-camera or hybrid spectral imaging systems (e.g., those using RGB+NIR or rotating filters). Without head-to-head comparisons to recent mobile spectral imaging works (like MobiSpectral, 2023), it is difficult to assess the true performance advantage of the proposed hardware design in practical conditions.

Also, I suggest the authors to compare with the CASSI system and the methods that is developed for the CASSI system.

(4) The improvements on the model design are not very significant.

**Questions:**

In Section 3.2, could you clarify how much the reconstruction performance would change if the corrected (optimal) filter pair were used, since the dataset was collected with a non-optimal one?

---

### Official Review · Reviewer_1jz6 · 2025-10-31

**Soundness:** 3
**Presentation:** 3
**Contribution:** 3
**Rating:** 4
**Confidence:** 3

**Summary:**

This paper proposes a multi-image hyperspectral reconstruction (MI-HSR) framework that leverages a triple-camera smartphone system with external spectral filters to improve hyperspectral image reconstruction from RGB images. The authors introduce the Doomer dataset, comprising 143 scenes captured with three smartphone cameras (two with custom filters) and a hyperspectral reference camera. They propose MI-MSFN, a modified transformer-based architecture that addresses spatial misalignment through spatial-first attention mechanisms. The system demonstrates approximately 30% improvement in spectral accuracy compared to single-camera RGB reconstruction.

**Strengths:**

1. The use of external filters on commodity smartphone cameras is creative and deployable without requiring custom hardware modifications. This represents a practical path toward accessible hyperspectral imaging.
2. The Doomer dataset fills an important gap by providing real multi-view RGB images with misalignment, spectral filters, and in-scene gray references—features absent in existing datasets. The detailed capture setup and preprocessing pipeline are well-documented.
3. The paper includes proper ablation studies, multiple baselines, statistical significance testing (10 runs with different seeds), and both simulated and real-world experiments. The comparison of aligned vs. original GT metrics adds transparency.

**Weaknesses:**

1. With only 143 scenes, the Doomer dataset is relatively small compared to other HSR datasets (e.g., ARAD_1K with 950 scenes). The limited diversity may affect model generalization and robustness evaluation.
2. The MI-MSFN architecture is primarily a reordering of existing MSFN components (spatial-first vs. spectral-first). While justified for the misalignment problem, the novelty is incremental. The improvement over MSFN is relatively small (Table 3: 29.86 vs 29.59 dB PSNR).
3. No comparison with other multi-view or multi-spectral reconstruction methods beyond RGB-only baselines. Comparison with methods like dual-camera super-resolution adapted to HSR would strengthen the evaluation.
4. The trained models are tightly coupled to specific hardware (Section 7), requiring new data collection and retraining for different camera systems. This significantly limits practical deployment despite the claimed accessibility.

**Questions:**

1. Have you explored or can you discuss potential strategies (e.g., transfer learning, domain adaptation) to reduce the retraining burden when changing camera systems?
2. Modern smartphones capture video. Have you considered extending this approach to video HSR, which could leverage temporal information for better reconstruction?

---

### Note · Authors · 2025-11-21

**Comment:**

We sincerely appreciate the reviewers’ insightful comments and constructive suggestions, which we will take into account in future revisions of our work

**Withdrawal Confirmation:**

I have read and agree with the venue's withdrawal policy on behalf of myself and my co-authors.